# Reprintable Paste-Based Materials for Additive Manufacturing in a Circular Economy

**Marita Sauerwein** [1,*], **Jure Zlopasa** [2], **Zjenja Doubrovski** [1], **Conny Bakker** [1] and **Ruud Balkenende** [1]

[1] Industrial Design Engineering, Delft University of Technology, 2628 CD Delft, the Netherlands; e.l.doubrovski@tudelft.nl (Z.D.); c.a.bakker@tudelft.nl (C.B.); A.R.Balkenende@tudelft.nl (R.B.)

[2] Applied Sciences, Delft University of Technology, 2628 CD Delft, the Netherlands; J.Zlopasa-1@tudelft.nl

[*] Correspondence: m.sauerwein@tudelft.nl

**Abstract:** The circular economy requires high-value material recovery to enable multiple product lifecycles. This implies the need for additive manufacturing to focus on the development and use of low-impact materials that, after product use, can be reconstituted to their original properties in terms of printability and functionality. We therefore investigated reprintable materials, made from bio-based resources. In order to equally consider material properties and recovery during development, we took a design approach to material development. In this way, the full material and product life cycle was studied, including multiple recovery steps. We applied this method to the development of a reprintable bio-based composite material for extrusion paste printing. This material is derived from natural and abundant resources, i.e., ground mussel shells and alginate. the alginate in the printing paste is ionically cross-linked after printing to create a water-resistant material. This reaction can be reversed to retain a printable paste. We studied paste composition, printability and material properties and 3D printed a design prototype. Alginate as a binder shows good printing and reprinting behaviour, as well as promising material properties. It thus demonstrates the concept of reprintable materials.

**Keywords:** product design; additive manufacturing; circular economy; bio-based resources; material integrity; recycling

## 1. Introduction

The circular economy is currently gaining momentum as it is viewed as a promising approach towards sustainable development. In a circular economy, products and materials are kept at their highest value for as long as possible by looping them back into the economy through reuse and recycling [1–3]. To achieve high-value and high-quality recovery, a shift of focus is required when it comes to product and material development. Instead of allowing products and materials to degrade and be wasted, they should be recoverable, reusable, and recyclable to enable the next lifecycle [4,5]. the material choice should therefore be based on the material properties as well as recovery options [6,7].

New design solutions with additive manufacturing (AM) or 3D printing provide opportunities for product life extension, reuse, and recovery in the circular economy. AM is a digital production process that, for example, allows for adjusting products to changing needs and contexts, thereby facilitating product life extension [8]. One of the issues with AM, however, is the limited availability of materials that can be recovered and reused at the end of a product's lifecycle, and thus complying with the aim of the circular economy. a typical example is Polylactic acid (PLA), one of the most commonly used materials for extrusion printing. Quality conservation after recycling is problematic as reheating reduces the material's rheological properties [9]. Other polymers, like Polyethylene

terephthalate (PET), are better suited to recycling, but these are based on non-renewable and oil-based resources [10]. Ideally, after a 3D printed product becomes obsolete, the material should be reprocessed into ready-to-print material which matches the original specifications with respect to print properties as well as functional properties.

Material sourcing is another important aspect. To comply with circular economy principles, we need to develop materials derived from bio-based and abundant resources as an alternative for oil-based materials [1,6]. Bio-based sources for AM are mainly studied in the field of bioprinting for tissue engineering. In this field, natural hydrogels like alginate, collagen, gelatin, and chitosan are commonly used [11]. the interest in bio-based sources for AM is currently expanding beyond this field, and is also being explored in product design. Faludi et al. [12], for example, printed products using bio-based paste materials to quantify the print energy and explore market viability. Sanandiya et al. [13] 3D printed a small wind turbine blade with a paste material from cellulose fibres bonded by chitin. Mogas-Soldevila et al. [14] created 3D print paste materials from chitosan for printing functionally graded materials. Tenhunen et al. [15] made a paste material based on cellulose for printing on textiles, and Rael and San Fratello [16] 3D printed a pavilion from salt and a natural glue using the AM method of binder jetting.

Makerspaces are an important driver for developments in AM; these are shared community workshops found all over the world [17]. These spaces enhance local innovation through, for example, the use of local material resources [18]. They give access to a wide variety of tools and machines to complete do-it-yourself making and digital fabrication projects [19]. the maker community is environmentally aware, but often lacks concrete guidance [20]. There is interest in guidance about sustainable materials within this community, as shown by, amongst others, Materiom, an online platform which provides open source recipes for materials from natural ingredients [21]. the increased interest in materials from natural and local (waste) sources is also evident in the design community. Designers are experimenting with materials such as coffee grounds, citrus peel, and agricultural residues [22,23] and developing their own custom-made design materials such as flip-flops from palm leather [24], tableware from ZandGlas [25], and 3D printed bowls and vases from seaweed filament [26].

Makerspaces and design communities are an attractive starting point for developing circular economy materials. In contrast to common material development processes that tend to focus on the optimisation of functional properties and manufacturability, a design approach permits quick and iterative testing of multiple lifecycle stages, from material sourcing to the end product. Moreover, it enables recovery options for a material in the early development stage. In this study, we therefore explore the complete lifecycle during material development, in order to create a proof of concept for reprintable materials. These can be fully reconstituted to their original specifications with respect to print and functional properties during use. This in turn enables high-value and high-quality material recovery.

In a previous study, we used this approach to developed a calcium carbonate-based material to explore reprintable materials for extrusion paste printing from bio-based and abundant resources [27]. We described the initial development of a composite material for paste printing with filler particles made from ground mussel shells and sugar water as binder material. the reprintability of this material was achieved through dissolving; after the print was air dried to obtain the final object, the object could be turned into a printable paste again through immersion in water. a lampshade was 3D printed to demonstrate the use of this material in a design object (Figure 1). However, an important disadvantage of this material is that it is inherently not water-resistant, which limits its application in product design.

In this study, we build on our preliminary study and specifically explore alginate as a binder for regenerative 3D paste printing. Alginate is a polysaccharide often used as a hydrogel in bioprinting because it increases the viscosity of water. the viscosity of alginate hydrogels is influenced by the molecular weight, the ratio of β-ᴅ-mannuronate (M) to α-ʟ-guluronate (G) blocks, and the concentration [11,28]. We use sodium alginate as it can be reversibly cross-linked through ion exchange, which provides an opportunity to achieve reprintability. We describe the process of

developing a reprintable paste material with this binder and a filler from ground mussel shells. Mussels shells are an abundant bio-based waste stream in the Netherlands. We present data about the paste composition, printability, material properties, and reprintability. Data for the material properties are obtained through tinkering (exploratory evaluation) to obtain a basic understanding of the material, mechanical testing to determine the technical characteristics, and from testing with participants to explore the material experience. In addition, a variety of fillers with sodium alginate was tested for material composition and printability and reprintability to explore the influence of different fillers on sodium alginate as a binder for 3D printing. Finally, we designed and 3D printed a prototype using mussel shell–alginate material, also from regenerated paste, to express the material characteristics and demonstrate the material properties in an actual application.

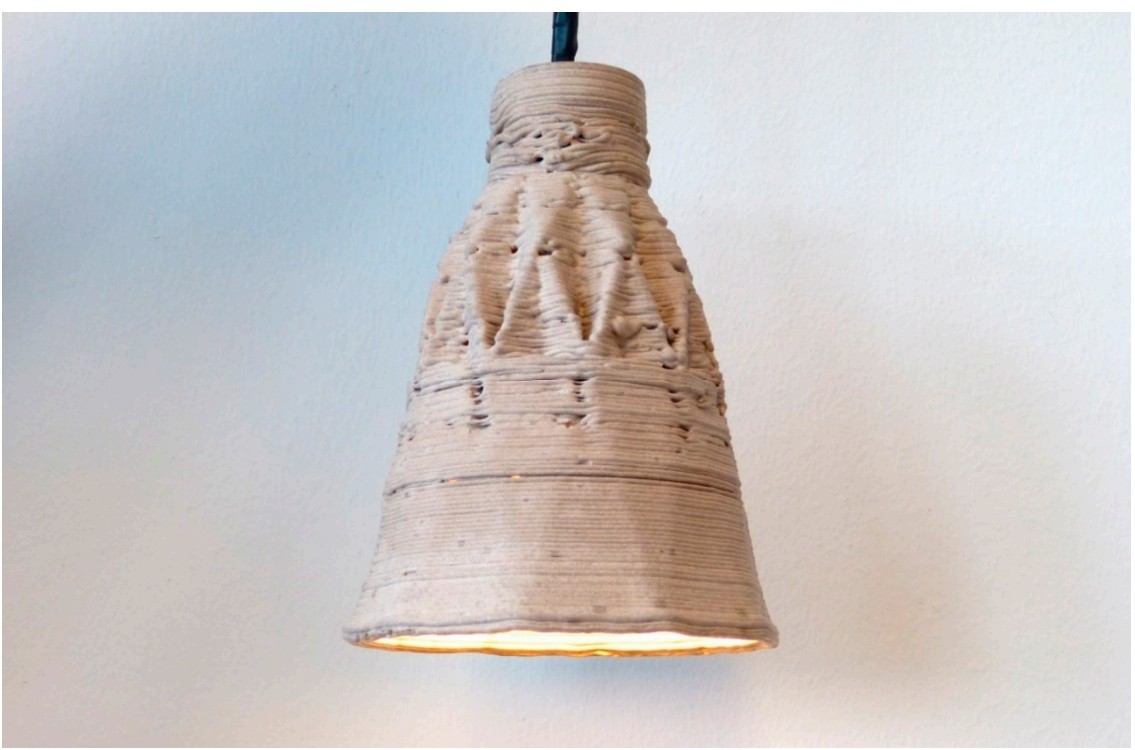

**Figure 1.** Lampshade 3D printed with a composite material from mussel shells and sugar water (design from Joost Vette).

## 2. Materials and Methods

### 2.1. Paste Materials

Water-based paste was developed with alginate as binder and filler particles derived from ground and sieved natural materials, primarily mussel shells. the reversibility of the printed material depends on its ability to re-dissolve the binder material. For binder material, we used sodium alginate as starting point (Sigma-Aldrich, Zwijndrecht, the Netherlands; used as received). This is a linear polysaccharide block copolymer of 1,4-linked β-ᴅ-mannuronate (M) and α-ʟ-guluronate (G) monomers with an average molecular weight of 150 kg/mol and an M/G ratio of 1.56. the sodium alginate was dissolved in tap water at room temperature to create the binder solution.

The opportunity to achieve reprintability is based on the formation of reversible ionic cross-links in the polymer binder. the initial printing paste was made using a water-soluble (non-cross-linked) sodium alginate that, after printing, was ion exchanged with a divalent cation, calcium. the calcium ion binds to two carboxylate groups of the polymer chains and forms a physical cross-link; this is commonly used to obtain rapid gelation [11,29,30]. We used this exchange reaction to make the dried

alginate water insoluble in a reversible way. the egg box model, as shown in Figure 2, is a schematic representation of the bonding between the polymer chains (box) by the calcium ions (egg) [31].

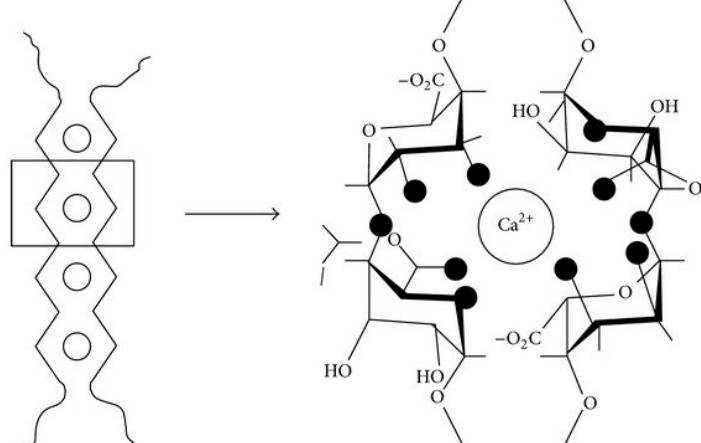

**Figure 2.** Schematic cross-linking of alginate with calcium ions according to the egg box model [2].

To retrieve a printable paste after use, the print is brought into contact with a sodium ion source. the calcium ions are then exchanged for sodium ions, thus reversing the cross-link between alginate chains to regain a soluble substance. Figure 3 gives an overview of the material reprintability process based on ion cross-linking.

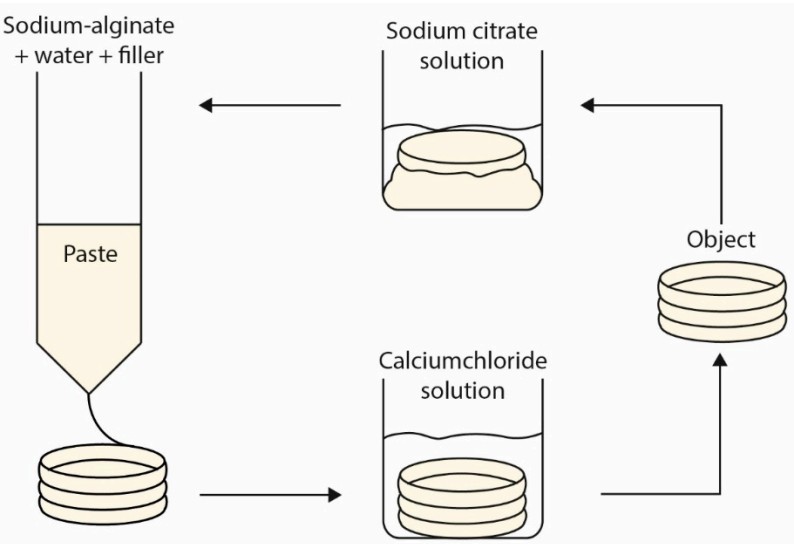

**Figure 3.** Reprintability process based on ion cross-linking.

For the majority of experiments, locally obtained ground mussel shells were used as filler material. the shells were collected at a Dutch processing plant in the province of Zeeland. Mussel shells are a large waste stream in the Netherlands; approximately 50 million kg of mussels are annually harvested, of which 40 wt-% consists of the shell [32,33]. Mussel shells consist of 95–99 wt-% layers of calcium carbonate connected by an organic matrix of chitin and silk-like proteins, accounting for 1–5 wt-% [34,35]. the shells were boiled in water for 20 min and subsequently heated in an oven for one hour at 200 °C to make them brittle for grinding and to dry out organic residues [33]. the shells were ground in an industrial food processer. the ground powder was sieved to obtain a maximum particle size of 75 μm, suitable for the size of our printing syringe.

A larger variety of natural filler materials was used to test the generalisability of alginate as binder material, i.e., eggshell, walnut shell, olive pomace, cacao shell, and pine and maple sawdust. These are all bio-based waste products and therefore potentially interesting resources. the variation of fillers is interesting as it allows us to use different local rest streams as well as vary the mechanical properties. Explorations in this study focused on the printability and demonstrating the reversibility from print to paste for the variety of fillers. We treated the eggshells (like mussel shells, they are calcium carbonate based) similarly to the mussel shells before grinding; other materials were ground as received and subsequently sieved for a maximum particle size of 125 μm for eggshell and walnut shell, and 75 μm for olive pomace, cacao shell, and pine and maple sawdust. the concentrations and amounts of solution that were needed to retain the paste for the different fillers were determined empirically.

### 2.2. Paste Composition

To create the paste for 3D printing, the sodium alginate was first mixed with water by hand stirring to obtain the binder solution. Subsequently, the filler material was added to this solution and mixed by hand, stirring until a homogeneous paste was obtained for printing. To obtain a printable paste, the viscosity needs to be sufficiently low to uniformly flow from the nozzle, and the paste needs to be stable enough to maintain its shape after printing. We used a syringe with the same nozzle as for 3D printing to deposit a paste track by applying pressure by hand to imitate extrusion printing. In this way, different ratios of binder, filler, and water could be rapidly tested. If this led to a satisfactory result, the paste was tested in the 3D printer. the table below outlines the weight ratios for the mussel shell–alginate material found to be suitable.

The composition of the paste from the other fillers was based on that of the mussel shell–alginate material (Table 1). It was varied until a printable paste was obtained for the specific filler; these combinations are shown in Table 2.

**Table 1.** Paste component weight percentages for mussel shell–alginate material.

|  | Sodium Alginate | Water | Mussel Shell Powder |
|---|---|---|---|
| Weight percentage | 3% | 36% | 61% |

**Table 2.** Paste component weight percentages for a variety of fillers.

| Fillers | Sodium Alginate | Water | Filler |
|---|---|---|---|
| Eggshell ≤ 125 μm | 6% | 40% | 54% |
| Walnut shell ≤ 125 μm | 4% | 75% | 21% |
| Olive pomace ≤ 75 μm | 4% | 66% | 30% |
| Pine sawdust ≤ 75 μm | 8% | 74% | 18% |
| Maple sawdust ≤ 75 μm | 7% | 78% | 15% |
| Cacao shell ≤ 75 μm | 5% | 66% | 29% |

After printing, the alginate samples were dried overnight at room temperature. the shrinkage during drying was determined by measuring the object's change in height and wall thickness.

### 2.3. 3D Printing Process

Test samples were made using an Ultimaker 2+ modified for paste printing with the Stoneflower Ceramic 3D Printing KIT Basic and micro printing set. In this system, the plunger of a syringe containing the paste is mechanically actuated with a stepper motor. a 60 cc syringe was used with a 14 gauge nozzle (1.6 mm inner diameter). the paste extrusion 3D printer was used in combination with the slicer software programme Cura to prepare the digital file for 3D printing. the layer height was set to 1.1 mm and the width to 1.5 mm, the print speed was set to 6 mm/s, and the extrusion speed to 0.19 mL/s.

### 2.4. Post-Treatment for Water Resistance

The alginate binder was made water insoluble by exchanging sodium ions with calcium ions. Although mussel shells mainly consist of calcium carbonate, the calcium ions in this filler do not cause any apparent cross-linking because of calcium carbonate's limited water solubility. Therefore, an external calcium source is needed to achieve gelation, for which we used calcium chloride ($CaCl_2$). For ionic cross-linking, the dried 3D printed object was submerged in a 2 wt-% calcium chloride dihydrate solution for 30 min (source: ≥99% purity, Sigma-Aldrich, the Netherlands; used as received). After removing the object from the solution, it was left to dry for several hours at room temperature until the object was no longer visibly damp.

### 2.5. Paste Regeneration

Sodium citrate attracts calcium ions and is commonly used as a buffer for ionic cross-linking of alginate [36]. We used a solution of water and trisodium citrate dihydrate (source: ≥99% purity, Sigma-Aldrich, the Netherlands; used as received) to regenerate the mussel shell–alginate printable paste from printed objects that were considered obsolete.

We first weighed the obsolete object(s) to determine the amount of sodium citrate and water. the solution was made from 2 wt-% trisodium citrate dihydrate and 50 wt-% water of the weight of the objects. the obsolete objects were then ground using a mortar and pestle to facilitate the dissolution process before adding the citrate solution. the mixture was stirred with a laboratory mixer until a homogenous and printable paste was achieved.

### 2.6. Technical Properties

The density, E-modulus, and flexural strength were determined to obtain an initial understanding of the technical properties of the ground mussel shell material. the density of the bio-based composite material was determined by measuring the weight and volume of printed rectangular bars. To measure the E-modulus and flexural strength, we performed a three-point flexural test according to ASTM International [37]. the test samples were 3D printed with a fully dense infill in the Cura slicing settings, and sanded afterwards to obtain precise dimensions of 3.0 by 4.0 by 40.0 mm with an accuracy of 0.05 mm. We then tested samples made from the virgin material, as well as those from the first and second reprinted material. Virgin samples were also tested in wet conditions after 30 min in water. the samples were tested in a Zwick Roell Z010 machine with a load cell of 500 N, tool radius of 10 mm, and test speed of 2 mm/min.

### 2.7. Experiential Properties

To obtain a basic understanding of the non-technical material properties, we tinkered with the material following Karana et al. [38]. the material's colour and smell were examined through direct observation. We tested brittleness by dropping a sample onto a hard surface from a height of approximately 1 m and flammability by holding it in a small flame for 10 s.

The Ma2E4 toolkit was used to test the experiential characteristics [39]. This toolkit provides guidance for testing a material sample on its performative, sensorial, interpretive, and affective qualities. the performative level explores different types of actions that the material causes, like touching, moving, and holding. the sensorial level describes several tactile and visual qualities, the interpretive level describes the meaning of the material, and the affective level the association with the material [39]. the shape of the test sample should be functionless (i.e., not implicating a direct function) as this might influence the perceived experience. Figure 4 shows the test sample. Twenty participants tested the experiential characteristics of the mussel shell–alginate material (male: 10, female: 10, age: 20–60 years, average age: 28, students from industrial design engineering: 13).

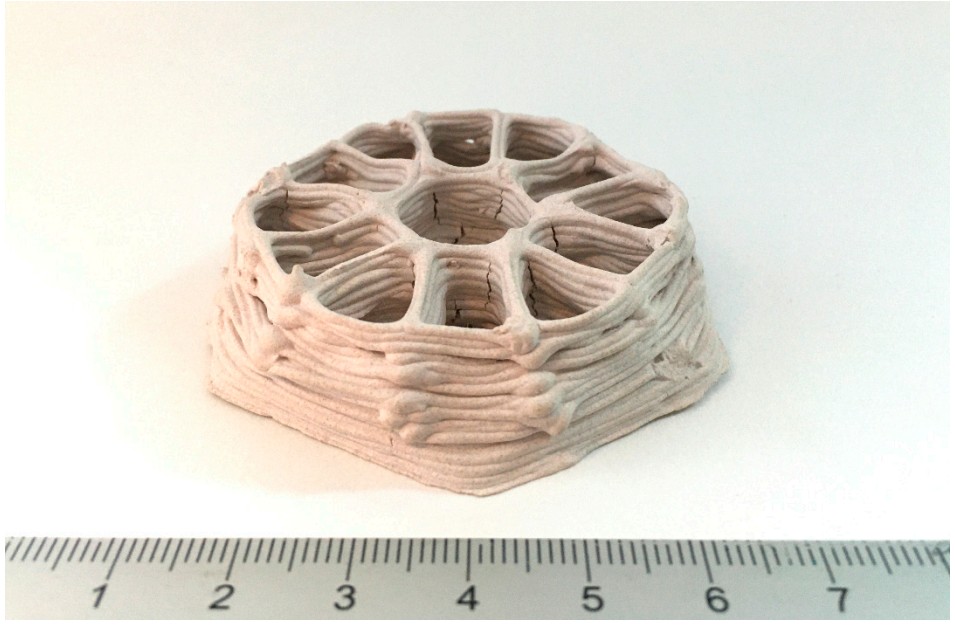

**Figure 4.** Sample to test the material experience using the Ma2E4 toolkit (design by Joost Vette).

*2.8. Prototype*

In addition to test samples, a prototype product was also designed and 3D printed to demonstrate the material properties in actual applications for the material based on the experiential and technical properties. a hair pin was developed as prototype. the design takes advantages of special properties of the mussel shell material that simplifies the 3D printing process and was used to explore experiential properties.

## 3. Results

*3.1. Technical Properties*

Table 3 presents the results of the flexural test and the density of the mussel–alginate samples. After breaking, internal cavities were visible in some test beams, demonstrating that the 3D printing process did not produce fully solid parts. However, the test results did not show a significant deviation for these samples in comparison to the homogenous test beams, indicating that the force at which fracture initiates was not determined by these flaws. the samples for the test with the wet virgin material were printed in two separate batches with significantly different treatments, which also affected the test results. the samples of batch 1 were printed with a less viscous paste (38% water) and dried for 44 days. the samples of batch 2 were dried for 3 days. Compared to the original material, the wet material in both cases exhibited a decrease of about three orders of magnitude in the flexural modulus and more than one order of magnitude in the E-modulus. This clearly shows the transition from a rigid material when dry to a ductile material when wet.

*3.2. Experiential Properties*

The mussel shell alginate material was off-white after printing and became whiter after drying. the smell was negligible. When dropped onto a hard surface from 1 m, the material samples broke. the cracks were clean cut, perpendicular to the printing lines. When held in a small flame, the material did not ignite, but burn marks were visible.

**Table 3.** Technical characteristics of the mussel–alginate composite material.

| Technical Properties | Virgin Material | Reprinted Material (1×) | Reprinted Material (2×) | Wet Virgin Batch 1 | Wet Virgin Batch 2 |
|---|---|---|---|---|---|
| Number of samples | 10 | 10 | 7 | 5 | 5 |
| Duration of paste drying (days) | 7 | 7 | 7 | 44 | 3 |
| Density (kg/m$^3$) | 1380 ($\sigma = 20$) | 1410 ($\sigma = 30$) | 1410 ($\sigma = 20$) | not determined | not determined |
| E-modulus (GPa) | 2.1 ($\sigma = 0.4$) | 2.0 ($\sigma = 0.4$) | 2.6 ($\sigma = 0.6$) | 0.0042 ($\sigma = 0.0005$) | 0.0014 ($\sigma = 0.0007$) |
| Flexural strength (MPa) | 9.8 ($\sigma = 0.8$) | 6.4 ($\sigma = 1.4$) | 6.9 ($\sigma = 1.9$) | 0.25 ($\sigma = 0.04$) | 0.08 ($\sigma = 0.01$) |

The results of the Ma2E4 toolkit are shown in Figure 5 and give an indication of the appearance of the material on the experiential levels, i.e., the performative, sensorial, affective, and interpretive level. It should be noted that the material appreciation may have been influenced by the differences in printing quality as well as it showing cracks after drying.

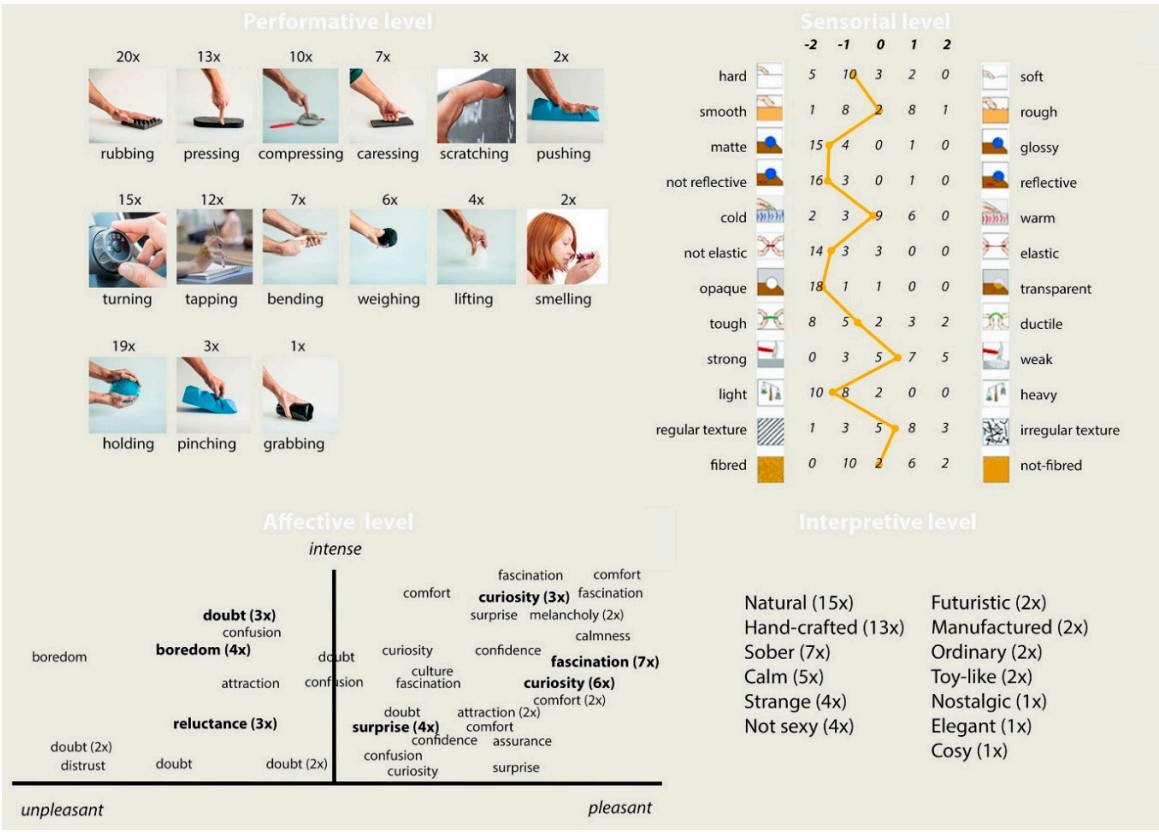

**Figure 5.** Test results of Ma2E4 toolkit.

On a performative level, the participants handled the material with care. They held it in their hands and rubbed it to feel the texture. the participants judged the material on a sensorial level to be lightweight and relatively weak and hard, as well as matt, non-reflective, non-elastic, and opaque. On the affective level, most words used were in the pleasant spectrum of the scale, of which fascination, curiosity, and surprise were mentioned most often. the participants also doubted about the material, noting characteristics like reluctance and boredom. Finally, the material was interpreted as being natural and hand-crafted, as well as sober and calm. Natural, hand-crafted, and calm had a positive connotation due to the material's uniqueness, light weight, and smooth surface. Sober had a negative connotation, because of the dull colour and matt finish.

### 3.3. Filler Variation

We tested the printability and reprintability of pastes with different fillers to explore the potential for filler exchange. the pastes were adapted with respect to the mussel shell recipe to achieve printability. Subsequently, all pastes were 3D printed using the same digital file so that the printed samples all exhibited the same geometry. Only the pine sawdust sample did not reach full height due to a too high viscosity of the paste, causing the material to slip behind the plunger. the drying of the 3D printed samples caused significant shrinkage in the case of the organic filler materials, as visible in Figure 6 and Table 4. Whereas shrinkage was only a few percent for the $CaCO_3$-based particles (mussel and eggshell), it varied from 10–40% for walnut, olive, pine, maple, and cacao.

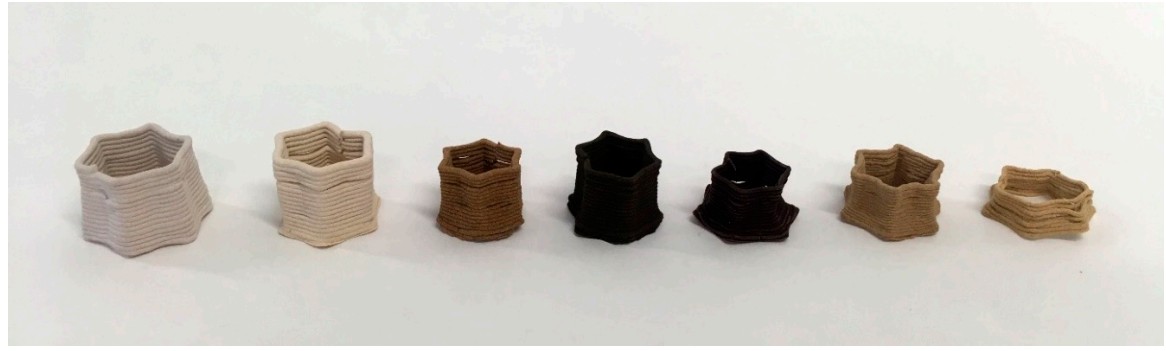

**Figure 6.** 3D prints with a variety of fillers, from left to right: mussel shell, eggshell, walnut shell, olive pomace, cacao shell, maple sawdust, and pine sawdust.

**Table 4.** Shrinkage percentage and results of drop test for all fillers.

| Filler | Shrinkage in % | | Breaks after Drop Test |
| --- | --- | --- | --- |
| | Height | Line Width | |
| Mussel shell | 4 | 6 | Yes |
| Eggshell 125 μm | 0 | 0 | Yes |
| Walnut shell | 26 | 12 | No |
| Olive pomace | 23 | 24 | No |
| Pine sawdust | 31 | 41 | No |
| Maple sawdust | 43 | 18 | No |
| Cacao shell | 35 | 41 | No |

Most samples felt rougher than the composite material prepared with ground mussel shells, only the composite material with cacao felt smoother. All materials could be broken by hand, but some did not break after dropping from approximately 1 m, as shown in Table 4. All materials were rigid when dry, became ductile when submerged in water, and could be returned to a paste after submersion in empirically determined aqueous trisodium citrate dihydrate solutions.

### 3.4. Prototypes

A hairpin was designed based on the technical and experiential characteristics of the mussel shell–alginate material. the material was experienced as light and had a soft touch, which made it pleasant to wear on the head. the brittleness in combination with the natural and ceramic look indicated a delicate material which was considered a good fit for hair decoration purposes. the 3D printing process could be simplified due to the unique property of the mussel shell–alginate material of being rigid when dry and ductile after submerging in water. the hairpin was printed flat (Figure 7) from virgin material and ion cross-linked with calcium chloride after drying. Subsequently, the hairpin was submerged in water to make the material ductile and slightly bent. It was left to dry in this position to obtain an end product that follows the contour of the head due to the curved shape (Figure 8). a second hairpin was printed using paste that was obtained for the third time after subsequent prints.

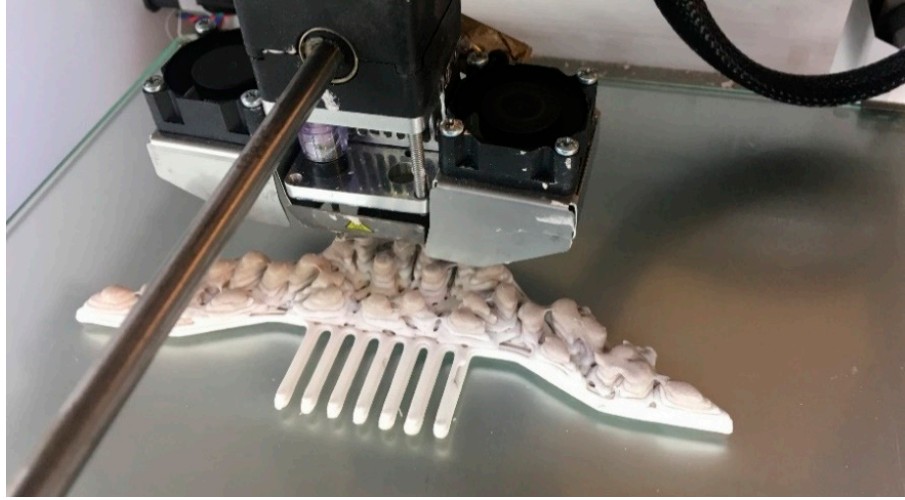

**Figure 7.** Flat 3D print of hairpin.

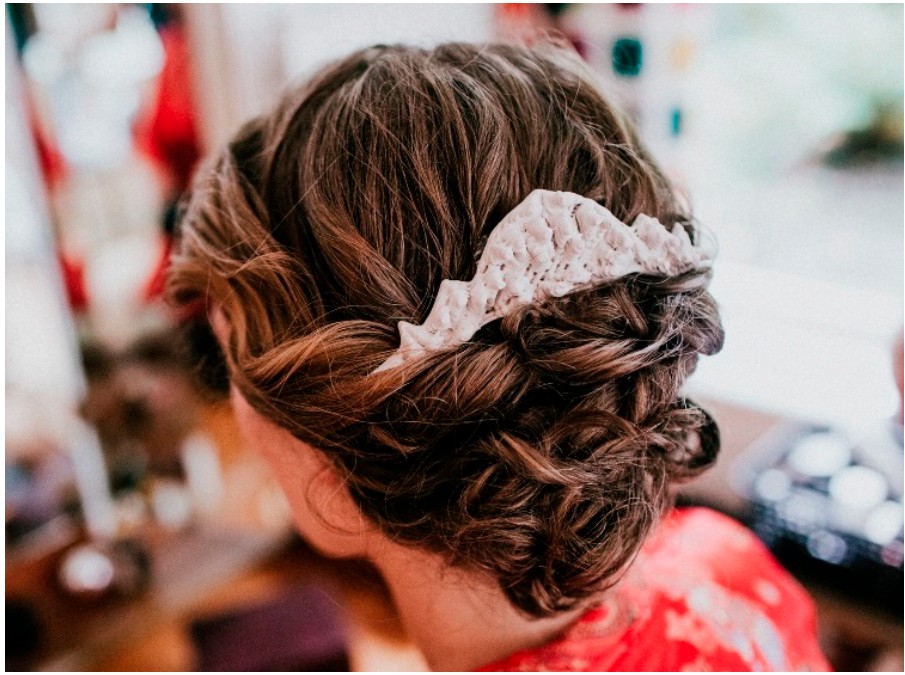

**Figure 8.** Final result of bended hair pin.

## 4. Discussion

We aimed to demonstrate a design approach to the development of materials for AM based on abundant natural resources that retain functional properties for reprinting after initial use. This supports the increased interest in developing AM materials from bio-based resources, and provides an explicit focus on end-of-life behaviour. From a circular economy perspective, the ability to recover the material at the end of product life is essential, so we explored the complete lifecycle of material development of reprintable paste-based materials using an iterative design approach. the goal of this broad and explorative approach was to develop a proof of concept; the material is thus not fully optimised from an engineering perspective.

Reprintable materials require a system perspective, as aspects ranging from 3D printability and characteristics for use and recovery have to be accounted for. By adopting a design approach, we were able to perform multiple iterations and consider the complete material lifecycle. We demonstrated this approach for the mussel shell–alginate material, making use of reversible ion cross-linking.

This resulted in a proof of concept that demonstrates the desired capacities as well as allowing for improvements and further optimisation. This approach provides insights into and guidelines for the first steps in material development that considers efficient reuse in addition to fabrication and use.

Although our aim was to fully reconstitute to the original material specification, the starting point for the reprinting material is different from that of the virgin material. Instead of mixing the pure starting materials to obtain a "virgin" paste, the printed object needs to be reprocessed again before reprinting. This reprocessing should have a minimal effect on the constitution and properties of the resulting paste. Dissolving the binder in water, as explored for a sugar binder [27], cannot be used after ion cross-linking because it results in water resistance. Therefore, in addition to water, a compound is needed to reverse the ion cross-linking. In this case, trisodium citrate dihydrate was added to exchange calcium ions with sodium ions, making the alginate soluble in a low environmental impact solution. the amount of trisodium citrate is negligible in theory, as a ratio of 1:0.0036 is needed to retain a printable paste with sodium alginate based on the molar masses of calcium alginate and trisodium citrate dihydrate. We added trisodium citrate dihydrate in a weight ratio to calcium alginate of 0.4:1 to obtain an acceptable speed of the dissolution process. This is a 100-fold excess compared to the stoichiometrically required amount, implying that in subsequent reprinting runs, some accumulation of citrate can be expected. Although this did not significantly affect the mechanical properties after two reprinting runs, the dissolution process may be subject to further optimisation.

The mechanical tests show that the properties of the dried material are largely maintained after a number of subsequent reprints with the original material. a slight decrease in the flexural strength was observed, but further experiments are needed to determine if this decrease is significant. Far more interesting is the 1000-fold decrease in flexural strength when, after cross-linking and drying, the printed object is returned to a wet state. As the material integrity is maintained, the achieved ductility allows for additional shaping operations, like the bending shown for the hairpin. the current investigations demonstrate the potential of the developed materials, however, more work is needed to establish the durability of the material.

Variations in filler material lead to pastes with properties comparable to the mussel shell paste regarding printer settings and water solubility after reversed ion cross-linking. It should be noted that the pastes were not fully optimised with respect to printability and subsequent material properties, as demonstrating the versatility of the process was our main purpose. the original mussel shell paste recipe was therefore only modified to obtain a printable paste. However, most of the fillers have a different composition, resulting in different filler properties. From the water to filler ratios in Table 2, it is evident that pastes with organic filler material need a much higher water content to achieve printability. This is attributed to the swelling of the organic particles in water; this also explains the different drying behaviour. Whereas the inorganic shell-based pastes hardly showed shrinkage, the organic particles exhibited 10–40% shrinkage, sometimes anisotropically. This large shrinkage can also be explained by the swelling of the organic particles in the paste. the observed anisotropy in shrinking directions is likely to be due differences in the shape of the particles. Further research on such modified pastes is needed to optimise the composition and resulting properties.

Reprintable materials are needed to make AM suitable for a circular economy. For paste extrusion printing, as done in this study, the binder acts as the adhesive between the filler particles and thus influences the reprintability, in addition to the printing and material properties. Its dissolution properties determine the approach to retaining a printable paste. Ion exchange to establish reversible cross-linking has been shown to be a promising dissolution process for retrieving reprintable materials. Other suitable binders for ion cross-linking could be explored to extend this group of materials and obtain a larger range of binder properties. Interesting options are, for example, pectin, carboxymethyl cellulose, and guar gum [40–42]. Furthermore, alternative mechanisms that could enable reprintability might be tested. Potentially interesting opportunities are thermosensitive bio-based binders, such as kappa carrageenan, agarose, or schizophyllan [43–45] or binders sensitive to changing pH values, such as chitosan [11].

The integration of design tools into the material development process is a promising approach for the development of a new, reusable material for 3D printing. Testing the experimental properties provided insights into how the material is perceived and provided guidelines for further development towards use cases and user acceptance. the material research was further enhanced by developing 3D printed prototypes to create tangible objects that gave a better indication of the material's abilities. Moreover, the prototypes enabled us to demonstrate the properties of and experience with the material and thus contribute to acceptance by highlighting interesting characteristics.

## 5. Conclusions

Only a limited number of AM materials are currently available that meet the requirements of a circular economy, i.e., maintaining high-level material integrity after use and enabling multiple use cycles. Resources, material properties, and material reusability should be equally considered during material development. We followed a method inspired by the design and maker community approaches, in which we went through the full material and product lifecycle of material development in order to meet the requirements of all stages and create a proof of concept.

We demonstrate the explorative development of reprintable materials. These are 3D printable materials that can be reconstituted to their original properties in terms of printability and functionality. Reprintability is obtained by retaining control over binder dissolution properties. We specifically describe the development of a reprintable bio-based composite material for extrusion paste printing from natural and abundant resources: ground mussel shells as filler and alginate as binder. Using calcium ions, the alginate binder is ionically cross-linked to obtain an insoluble material which can be reversed to retain a printable paste. In addition to reprintability as a unique property, an interesting characteristic of this material is its rigidity in dry conditions, while being ductile and shapable after submersion in water.

Alginate as a binder shows good printing and reprinting behaviour, hence contributing to this new group of materials for AM. More material research is needed to further explore the properties of alginate-bonded composites and the durability, as well as other solutions for reprintability. It is essential to create a pallet of reprintable materials that maintain high-level integrity in a circular economy by equally considering material properties and recovery during development.

**Author Contributions:** Conceptualisation, M.S., Z.D., C.B., and R.B.; methodology, M.S., J.Z., Z.D., C.B., and R.B.; validation, M.S., J.Z., and R.B.; resources, J.Z.; data curation, M.S.; writing—original draft preparation, M.S.; writing—review and editing, J.Z., Z.D., C.B., and R.B.; visualization, M.S.; supervision, Z.D., C.B., and R.B. All authors have read and agreed to the published version of the manuscript.

**Funding:** This research received no external funding.

**Acknowledgments:** We would like to thank Joost Vette and Edwin van Tongeren for their contribution to the development of reprintable composite materials during their graduation work.

**Conflicts of Interest:** The authors declare no conflict of interest.

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
