# Peer review of "Reprintable Paste-Based Materials for Additive Manufacturing in a Circular Economy"

_sustainability, doi:10.3390/su12198032_

Round 1

Reviewer 1 Report

The manuscript adresses an interstign topic of reprintabel paste for additive manufacturing in a circualr economy. Nowadays 3D printing is a linear process with disposing at the end of the life time. There is nearly no attempt to reuse the material. This makes the paper highly interesting for the readers and should be published. It should also be highlighted, that the study did not receive any funding.

I still have a few comments that are listed below:

1) Why are mussel shells used? There was never given a reason for that. Did you use mussel shells as waste products? Where did you collect the material and how much material is available for printing? I can imagine that this process might work in areas close to the sea, but it won't work in other  areas and you have a lot of transportation emissions. Please also explain the origin of cacao shells, pine, etc.

2) Line 145-156: You missed to deleete this section as this was part of the format style.

3) Figure 4 and the chapter 2.8 is misleading. As the hair pin is mentioned, it was uncelar if figure 4 and the sample represents already the hair pin. Maybe you refer to the later picture 8 or make it otherwise clear that these are two different samples. As you have the same chapter as 3.4. I suggest to delete chapter 2.8. and explain the prototpye in chapter 3.4. Please also insert a scale into figure 4.

4) Line 335: a word seems to be missing in line 335 between: "the printed....is returned."

5) Line 337ff.: You state that variations in filler material show camaprabel results to the mussel shell paste. Where is the proof for that? We have only your opinion here. Are all pastes reversible?

6) I was missing totally the results of the other materials as cacao shell, pine etc. There were only mentioned in the experiment set us and there is no result shown.

As this paper adresses a new topic we need more papers in this direction to compare the results. There is still some work to be done.

I assume that my comments are easy to solve for you and therefore I recommend minor revisions.

Author Response

Dear reviewer,

Thank you for the constructive review. We edited the manuscript based on your comments. Changes in the document are indicated with ‘track changes’ and you can find our reaction on your comments below in red.

The manuscript adresses an interstign topic of reprintabel paste for additive manufacturing in a circualr economy. Nowadays 3D printing is a linear process with disposing at the end of the life time. There is nearly no attempt to reuse the material. This makes the paper highly interesting for the readers and should be published. It should also be highlighted, that the study did not receive any funding.

I still have a few comments that are listed below:

  • Why are mussel shells used? There was never given a reason for that. Did you use mussel shells as waste products? Where did you collect the material and how much material is available for printing? I can imagine that this process might work in areas close to the sea, but it won't work in other  areas and you have a lot of transportation emissions. Please also explain the origin of cacao shells, pine, etc.

A better clarification is given in line 135-137 and line 144-146

  • Line 145-156: You missed to deleete this section as this was part of the format style.

We deleted this section

  • Figure 4 and the chapter 2.8 is misleading. As the hair pin is mentioned, it was uncelar if figure 4 and the sample represents already the hair pin. Maybe you refer to the later picture 8 or make it otherwise clear that these are two different samples. As you have the same chapter as 3.4. I suggest to delete chapter 2.8. and explain the prototpye in chapter 3.4. Please also insert a scale into figure 4.

The prototyping process was part of the method to demonstrate an actual application, we therefore consider it appropriate to keep paragraph 2.8. However, we adapted the introductory wording in line 226 to avoid confusion. Also we replaced figure 4 by an image with scale bar.

  • Line 335: a word seems to be missing in line 335 between: "the printed....is returned."

We included the word ‘object’

  • Line 337ff.: You state that variations in filler material show camaprabel results to the mussel shell paste. Where is the proof for that? We have only your opinion here. Are all pastes reversible?

In Section 2.1 the aim and setup of the experiments with filler variation has been better explained (line 144-154). In section 3.3 the obtained result is described in  more detail.  

  • I was missing totally the results of the other materials as cacao shell, pine etc. There were only mentioned in the experiment set us and there is no result shown.

The results of the other fillers are described in section ‘3.3 filler variation’.  It should be noted that only printing and reversibility tests were done (this now clear from section 2.1) with these fillers and the mechanical properties were not determined.

As this paper adresses a new topic we need more papers in this direction to compare the results. There is still some work to be done.

I assume that my comments are easy to solve for you and therefore I recommend minor revisions.

Reviewer 2 Report

Overall

This paper presents the development of a bio-based reprintable material and its tests in additive manufacturing.

Abstract is clear and well structured, presenting all the elements needed (introduction to topic, methodology, findings).

Keywords could include “reprintable materials” instead of recycling, since it is the term most used in the document

It is the reviewer opinion that the paper is well written, in a clear manner, helpful to understand the subject. Is original and presents a step forward in the area.

The paper is structured in a simple but correct way, presenting a direct relation between the work developed and the paper chapters.

  1. Introduction

This chapter presents a clear state of the art on circular economy, additive manufacturing and materials, but lacks to point out the durability aspect of products and materials in the context of circular economy, since (besides closing loops) it is also intended to slow loops.

In the end of this chapter it’s unclear if a reprint was conducted.

  1. Materials and Methods

Line 145 to 156 should be removed

Table 2 caption should start with capital letter

Lines 193 to 200 – The obsolete objects are printed objects? It unclear… Please clarify what you mean by obsolete objects.

  1. Results

Chapt 3.4 It’s unclear if the material used in the first hair pin is virgin or is already reprinted. The last phrase “Using material recovered for the third time, we successfully 3D printed another hairpin.” mentions a third impression, but it’s unclear the sequence (1st, and 2nd prints)

  1. Discussion

Impact of compounds for material reprocessing should be discussed, so that the dissolution impacts can be weighted.

  1. Conclusions

This conclusion and recommendations are in line with what has been analyzed.

In the recommendations for further works it should be addressed the durability test of the material.

The paper presents a good literature review and references are up to date.

Author Response

Dear reviewer,

Thank you for the constructive review. We edited the manuscript based on your comments. Changes in the document are indicated with ‘track changes’ and you can find our reaction on your comments below in red.

This paper presents the development of a bio-based reprintable material and its tests in additive manufacturing.

Abstract is clear and well structured, presenting all the elements needed (introduction to topic, methodology, findings).

Keywords could include “reprintable materials” instead of recycling, since it is the term most used in the document.

We have chosen for ‘recycling’, because this is a more generic term and will therefore be more frequently used in searching. We consider our paper to be relevant in searches combinating ‘recycling’ and ‘3D printing’.  

It is the reviewer opinion that the paper is well written, in a clear manner, helpful to understand the subject. Is original and presents a step forward in the area.

The paper is structured in a simple but correct way, presenting a direct relation between the work developed and the paper chapters.

  1. Introduction

This chapter presents a clear state of the art on circular economy, additive manufacturing and materials, but lacks to point out the durability aspect of products and materials in the context of circular economy, since (besides closing loops) it is also intended to slow loops.

We agree on the importance of closing and slowing loops in a circular economy. The aim of this paper is to investigate closing of specific material loops without quality loss and does not intend to discuss all related circular economy aspects (that will need an additional study). Therefore, to address the importance of durability, we added a recommendation for further research on durability in the discussion (line 346-348) and conclusion (line 400). 

In the end of this chapter it’s unclear if a reprint was conducted.

Some adaptions have been made to the last paragraph of the introduction to clarify this

     2. Materials and Methods

Line 145 to 156 should be removed

We deleted this section

Table 2 caption should start with capital letter

Done

Lines 193 to 200 – The obsolete objects are printed objects? It unclear… Please clarify what you mean by obsolete objects.

A clarification is added in line 190-191

     3. Results

Chapt 3.4 It’s unclear if the material used in the first hair pin is virgin or is already reprinted. The last phrase “Using material recovered for the third time, we successfully 3D printed another hairpin.” mentions a third impression, but it’s unclear the sequence (1st, and 2nd prints)

A clarification is given in line 297 and 300-301

      4. Discussion

Impact of compounds for material reprocessing should be discussed, so that the dissolution impacts can be weighted.

We assume that the reviewer refers to environmental impact here. The dissolution substance consists of sodium citrate and water, which are both substances with a low environmental impact, which we added in line 332-333.

      5. Conclusions

This conclusion and recommendations are in line with what has been analyzed.

In the recommendations for further works it should be addressed the durability test of the material.

We now address durability in line 400

The paper presents a good literature review and references are up to date.
